# Which outcomes should always be measured in intervention studies for improving work participation for people with a health problem? An international multistakeholder Delphi study to develop a core outcome set for Work participation (COS for Work)

Margarita Ravinskaya ,[1] Jos H Verbeek,[1] Miranda Langendam,[2] Ira Madan,[3,4] Suzanne M.M. Verstappen,[5,6,7] Regina Kunz,[8] Carel T.J. Hulshof,[9] Jan L. Hoving,[1] Delphi participants

For numbered affiliations see end of article.

**Correspondence to**
Margarita Ravinskaya;
m.ravinskaya@amsterdamumc.nl

## ABSTRACT

**Objective** Synthesising evidence of the effects of interventions to improve work participation among people with health problems is currently difficult due to heterogeneity in outcome measurements. A core outcome set for work participation is needed.

**Study design and setting** Following the Core Outcome Measures in Effectiveness Trials methodology, we used a five-step approach to reach international multistakeholder consensus on a core outcome set for work participation. Five subgroups of stakeholders took part in two rounds of discussions and completed two Delphi voting rounds on 26 outcomes. A consensus of ≥80% determined core outcomes and 50%–80% consensus was required for candidate outcomes.

**Results** Fifty-eight stakeholders took part in the Delphi rounds. Core outcomes were: 'any type of employment including self-employment', 'proportion of workers that return to work after being absent because of illness' and 'time to return to work'. Ten candidate outcomes were proposed, among others: 'sustainable employment', 'work productivity' and 'workers' perception of return to work'.

**Conclusion** As a minimum, all studies evaluating the impact of interventions on work participation should include one employment outcome and two return to work outcomes if workers are on sick leave prior to the intervention.

## STRENGTHS AND LIMITATIONS OF THIS STUDY

⇒ The core outcome set was developed in accordance with the Core Outcome Measures in Effectiveness Trials guidelines.
⇒ A five-step approach was used to reach international multistakeholder consensus on a core outcome set for work participation.
⇒ Five key stakeholder groups were involved in the consensus process: researchers, occupational health professionals, policy makers, employee/patient representatives and an employer.
⇒ Provided with comprehensive background information the stakeholders participated in two group discussions to discuss perspectives following a transparent Delphi procedure.
⇒ The majority of the stakeholders were researchers from Europe and North America.

## INTRODUCTION

Authors of systematic reviews (SRs) state that inconsistent reporting of work participation outcomes in clinical trials hampers evidence synthesis in the field of occupational health (OH).[1–3] Coordinating editors of Cochrane review groups have indicated that the reliability and quality of SRs could be improved by using core outcome sets (COS).[4] In 2019, the Coronel Institute of OH at the Amsterdam University Medical Centre established an international research collaboration to develop a COS for Work Participation (COS for Work), based on the Core Outcome Measures in Effectiveness Trials (COMET) methodology.[5] Our aim was to develop a generic COS for Work to be used in intervention studies which expect to have an effect on work participation. As our COS is generic, and relevant to different diseases, it can be used for any type of intervention. For example, pharmaceutical studies for people with rheumatoid arthritis may not only improve clinical outcomes but indirectly also help participants

participating in work. Such studies mostly measure work outcomes as a secondary outcome. Other types of interventions aim directly in helping people with work participation, such as return to work interventions for cancer survivors or workers with long COVID.

The first phase of our project involved an SR on the spectrum of work outcomes used to measure the effect of interventions in published randomised controlled trials (RCTs).[6] The SR showed extensive heterogeneity in work-participation outcome measurements and confirmed the need for a COS for Work. In addition, we saw that it was unclear why authors chose to measure outcomes in a certain way and that to create COS for Work we would need to have a framework for outcomes which would ensure meaningful, pragmatic and mutually exclusive categories in which outcomes could be placed. In the second phase, we created a framework to aid the selection of work participation outcomes.[7] The framework outlines four main stages of work participation: outcomes assessing whether a person is successful in acquiring or initiating employment (stage 1), outcomes indicating whether a person is in employment, can retain employment or lost work within the duration of the study, that is, having employment (stage 2), outcomes that measure increasing or maintaining productivity at work (stage 3) when persons experience limitations or restrictions with working or have less output, and outcomes that assess return to employment and sick leave from work (stage 4) when people (temporarily) stop attending work and are on sick leave. Phase three involved a survey among reviewers and trial authors showing different reasons and preferences for choosing and using work outcomes. In particular, researchers choose outcomes because of their use in similar studies or their relevance for the health problem or type of intervention.[8]

In the fourth phase, we sought international consensus on COS for Work via a Delphi study. In the fifth and final phase, we will make recommendations for clinimetrically sound methods to measure the core outcomes selected during phase 4.

The overall aim of the Delphi study is to define a comprehensive and minimal set of outcomes that is relevant and feasible for measuring the effectiveness of interventions that result in a change in work participation, based on consensus of an international, multistakeholder group.

## METHODS
### On-line multistakeholder discussions and Delphi voting
To develop methods, we followed the COMET guidelines,[5] that is, the COS-STAndards for Development (The COS-STAD).[9] For reporting we used the COS-STAndards for reporting (COS-STAR).[10] The protocol for this study was published on the COMET website prior to the commencement of the study (https://www.comet-initiative.org/studies/details/1195, online supplemental file 1), and includes an extensive explanation of the methodology. The five steps are summarised briefly below.

### Step 1: preparation
We invited five groups of stakeholder representatives:
1. OH researchers—experienced researchers in OH field from varying disciplines, such as: health economy, occupational mental health, insurance medicine, workplace interventions, RtW interventions.
2. OH professionals—physiotherapists, orthopaedic surgeon, rehabilitation specialist.
3. Workers/employees and patients—stakeholders representing patients and workers as members of patients federations, workers alliances and labour unions.
4. Policy makers—stakeholders working for organisations creating OH policies such as the National Institute for Insurance against Accidents at Work, European Agency for Safety and Health at Work.
5. Employer—OH case manager.

These five stakeholder groups were chosen as key stakeholders. Researchers will use the COS for Work and have experience with work participation outcomes measurement and the remaining stakeholder groups will use the results of what is researched in their professional or daily life.

Stakeholders were recruited from our phase two international survey[8] and our professional network. We anticipated that an effective group discussion with up to 30 participants would be feasible to allow diversity of opinion in the various subgroups. Our aim was to organise two group discussions of 30 participants, in the morning and late afternoon to account for the varying time zones. We sent out 90 invitations.

Stakeholders who agreed to participate received an information package that included: (1) instructions for Delphi participation (online supplementary file 2), (2) a document containing the outcomes that would be included in the Delphi with worksheets to prepare for the first discussion (table 1, online supplementary file 3) and (3) a list of all participants in each discussion group. Item generation for the 24 outcomes in the preliminary set (table 1) was based on the SR[6] and categorised under the 4 work participation stages as described in the framework (table 1).

Participants were invited to familiarise themselves with the material and add any outcomes, which they deemed missing from the preliminary list prior to the first discussion (online supplementary file 4). The research team evaluated whether these new outcomes fitted the COS for Work criteria (table 2) and presented the outcomes during the first discussion round.

### Step 2: first online consensus stakeholder meeting
In the first online meeting, we presented the steps and research findings of the COS development including the four stages of work participation (see table 1) and criteria for COS outcomes (table 2). Further, we elaborated on the various stakeholder perspectives and its effect on

**Table 1** The preliminary set of outcomes that was presented to the stakeholders prior to the first discussion

| Stage of work participation—interventions and study populations for which outcomes are relevant within the stage | Outcome to be voted on during the Delphi |
|---|---|
| Stage 1: Initiating employment: Interventions aiming to help unemployed people with distance to the job market due to a health problem get work | 1. Skills for job procurement. Total work-related network, engagement in work seeking activities, job interview skills, work-related benefit of vocational training |
| | 2. Self-efficacy. Self-efficacy for job procurement, hope, optimism and self-efficacy for achieving vocational success, social/interpersonal self-efficacy |
| Stage 2: Having employment Relevant for any intervention and any type of health problem. Any type of health condition may impair work participation at the most elemental level, having work | 3. Employment. Having any type of employment including self-employment |
| | 4. Employment. Having met a predefined status of employment; part/full time, competitive work in a mainstream setting, performing the same tasks as non-disabled workers |
| | 5. Employment. Predefined type or amount of income |
| | 6. Employment. Duration of employment |
| | 7. Employment. Having lost employment |
| | 8. Work disability. Permanent and complete inability to engage in any work participation |
| | 9. Work disability. Permanent partial disability to engage in work participation |
| Stage 3: Increasing or maintaining productivity at work Interventions expected to help workers who are not on sick leave to maintain or increase productivity at work despite a health problem | 10. Work productivity loss (economic evaluation). Overall loss of productivity at work (in terms of quality or output at work) resulting from presenteeism or absenteeism regardless of cause. Typically used for economic evaluations |
| | 11. Presenteeism. Not being productive while at work due to health problems. Typically used to evaluate the effect of health interventions for individuals |
| | 12. Work ability. Current self-rated physical and mental work ability compared with lifetime best. |
| | 13. Work activity impairment. Experienced functional impairment of performing work activities due to a health problem |
| | 14. Perceptions affecting productivity. Possible constructs: beliefs in working capacity, motivation for work, vocational commitment, job coping, contentment with work, need for recovery |
| Stage 4: Return to employment and sick leave from work Return to work: Interventions expected to help people who are on sick leave return to work Sick leave from work: 1. Interventions including people at risk of (frequent) sick leave due to a health problem, for example, chronic conditions or recovery after an injury or procedure 2. Interventions aiming to prevent overall sickness absence among all workers | 15. Return to work—proportion of workers that return to work after being absent because of illness |
| | 16. Return to work—time to return to work or duration of sick leave after being absent because of illness |
| | 17. Sustainable return to work. Proportion of workers that return to work and remain at work for a specified amount of time |
| | 18. Sustainable return to work. Time to return to work in workers absent because of illness and who do not relapse within a specified amount of time |
| | 19. Return to work—the worker's perceptions. Time it will take to return to work as perceived by the sick worker also described as self-efficacy for return to work, return to work expectations or intention to return to work |
| | 20. Sickness absence among all workers in an organisation/ group expressed as lost working days or lost sick days |
| | 21. Sick leave duration. Working days absent because of illness averaged over all workers |
| | 22. Sick leave frequency. No of episodes of sick leave per worker expressed as percentage |
| | 23. Sickness absence among absent workers in an organisation/group expressed as lost working days or lost sick days |
| | 24. Sick leave duration. Working days absent because of illness averaged over all workers |

Item generation was based on the systematic review on how work participation outcomes are currently measured. Online supplemental file 3 contains the full overview with examples of how the outcomes could be measured.

outcome prioritisation, and discussed the proposed additional outcomes. In breakout rooms, groups of stakeholders created and discussed 'absolutely in' and 'absolutely out' outcomes for each stage. The results were discussed in the plenary session.

### Step 3: first round of Delphi voting
COMET online DelphiManager software participants voted on a total of 25 outcomes (outcomes from table 1 plus one outcome suggested by a stakeholder) accompanied by clarifying text on how it could be measured. With a personal login, the DelphiManager software allowed each outcome to be anonymously ranked on a Likert Scale from 1 to 9: from not important (0 points) to critical (nine points), including an 'unable to score' option. Participants were asked to provide rationale for their ratings and additional outcomes could be suggested. A

**Table 2** Criteria for COS outcomes which the stakeholders were asked to consider when deciding for or against inclusion of outcomes in COS for work

| Criteria to consider | Optional criteria |
|---|---|
| Outcomes should:<br>1. Be sensitive to change<br>2. Be feasible to measure<br>3. Be applicable internationally<br>4. Be work participation specific<br>5. Capture multiple stakeholder perspectives<br>6. Be in alignment with the International Classification of Functioning, Disability and Health model | Outcomes should if possible/applicable:<br>1. Be used for cost-effectiveness studies<br>2. Be applicable across varying insurance schemes |

COS, core outcome set.

single voting reminder was sent to participants 2 days before the closing of the first round.

### Step 4: second online stakeholder meeting

The results of the first online Delphi voting round and a collated list of arguments on why to include or exclude an outcomes were first distributed via email and then discussed in the second online stakeholder meeting. We also presented our analysis of suggested additional outcomes (online supplementary file 5). Based on the a priori defined consensus definition (see 'data analysis') and input from the stakeholder meeting, the authors then decided which outcomes would be dropped, included, reformulated or added to the second Delphi round.

### Step 5: second Delphi voting round

The same online Delphi procedure allowed each participant to individually rate the outcomes that remained from the first Delphi round. As before, reminders were sent to non-completers.

### Patient and public involvement

Consultation of patients' is seen as a crucial aspect in the development of COS.[5] We have consulted patient representatives, who were also representing the 'employee' perspective, in the discussion rounds and the selection of core outcomes for COS for Work. The patient representatives were members of, among others: The Netherlands Patients Federation, Canadian Injured Workers Alliance and labour unions. In addition, we also involved policy makers active in the field of OH. The results were disseminated to all participants via e-mail.

### Data analysis

After the first and second stakeholder meetings we held a meeting with the steering group to discuss how we could use and present the stakeholder input for the Delphi voting to inform all Delphi participants. The results from the Delphi rounds were extracted from the DelphiManager software into a comma-seperated file format and analysed in SPSS (version number 28) using descriptive statistics.

For each round, we calculated the distribution of scores for all outcomes as well as the percentage of stakeholder scoring between 7 and 9. As per protocol, outcomes scored between 7 and 9 by at least 80% of all stakeholders were deemed as core—showing as 'definite core outcome IN' on the spreadsheet distributed among stakeholders. Outcomes receiving a mean score between 7 and 9, from 50% to 80% of the participants in the first Delphi round were 'candidate outcomes' to be discussed with the stakeholders and rerated in the second Delphi round— labelled as 'possible core outcome IN'. Outcomes which received a score between 7 and 9 from less than 50% of the participants were dropped after the first round. For outcomes that were not considered definite core outcome but for which there was strong disagreement between stakeholder groups in the first round, the steering group still considered them for being discussed and rated in the second round. It was possible to upgrade or downgrade the ranking of an outcome for inclusion or exclusion in the second Delphi round based on argumentation of stakeholders. The final COS is based on the rankings of the second round.

## RESULTS

### Respondents

Initially, we approached 90 potential stakeholders and 59 stakeholders agreed to participate (not everyone from this group registered for the Delphi). It was difficult to recruit employer and employee representatives; we had no replies from employee representatives and one employer. After the first discussion round, participants helped us to recruit 11 employee representatives through their professional network. In total 70 stakeholders agreed to participate. Twelve participants took part in either one or two of discussions but did not register for the Delphi-Manager software, some of whom emailed that they did not manage to due to time constraints (see table 3 for participant characteristics).

### First online consensus stakeholder discussion

We presented all Delphi voting information (ratings and arguments) to the stakeholders as outlined in the methods section. The stakeholders discussed their preferences for and against the outcomes in four breakout rooms. They then presented their arguments in a plenary session that included discussion on topics such as:

**Table 3** Characteristics of survey respondents for two Delphi rounds (n; %)

| | Delphi round 1 No of participants | Delphi round 2 No of participants |
|---|---|---|
| Stakeholders | 58; 100 | 58; 100 |
| Researcher | 35; 60 | 37; 63 |
| Employee | 10; 17 | 10; 17 |
| Occupational health professional | 4; 7 | 4; 7 |
| Employer | 1; 2 | 1; 2 |
| Policy maker | 5; 9 | 3; 5 |
| Non-completers | 3; 5 | 3; 5 |
| Location of residence | | |
| Europe | 31; 53 | 30; 52 |
| North America | 17; 31 | 17; 29 |
| Australia | 3; 5 | 4; 7 |
| Asia | 3; 5 | 3; 5 |
| South America | 1; 2 | 1; 2 |

In total 58 participants registered as participants in the DelphiManager. Each participant belonged to one stakeholder group. In every round, three participants did not complete the voting.

► When work can be considered as sustainable and healthy and the challenges of measuring such concepts as part of this COS.
► Which types of outcomes would be applicable and feasible to measure in an international context.
► Why COS outcomes should have a positive focus (such as work-ability) rather than negative (such as sick leave).

We received fifteen outcomes (online supplementary file 4) which stakeholders found missing from the preliminary set (table 1). We did not include the outcomes which did not match with the COS criteria (table 2), outcomes which were already included but phrased slightly different, and outcomes which were a specification on the level of measurement. Two outcomes were added to the list for the first round of Delphi voting: (1) '(self)-perceived stigma' to the perceptions affecting productivity outcome concept and (2) 'time to employment' to the initiating employment stage.

### First round of Delphi voting and second online stakeholder discussion

The ratings for the first round of Delphi voting are presented in online supplementary file 6. None of the outcomes within the four stages received our cut-point for 'definite core outcome IN' (>80% score of 7–9 points).

### Stage 1: initiating employment

The highest-ranking outcome was 'time to employment' with a mean score of 5.6 and 44% of stakeholders giving it a score between 7 and 9. To not entirely exclude this stage

from the second round, the steering group included the newly proposed outcome 'any type of employment including self-employment' for the second Delphi round.

### Stage 2: having employment

'Any type of employment' was the highest-ranking outcome with a mean score of 7.4 and 75% of the stakeholders scoring it between 7 and 9. Stakeholders indicated that this outcome was feasible to measure internationally irrespective of social insurance schemes. In addition, this outcome was also seen as an indicator of the degree of work participation when sickness absence is not measured, and people lose work because of a health problem—relevant for countries where sickness benefits are limited. An additional outcome was proposed by a stakeholder, discussed and agreed for inclusion in the second round: 'sustainable employment' (online supplementary file 5). The employee representatives advocated to define 'sustainable' in terms of quality of the working life, such as having decent work and healthy work.

### Stage 3: increasing or maintaining productivity at work

'Work productivity loss', 'Work activity impairment' and 'Work ability' received 'possible core outcome IN' rankings, and during the discussion, there were no objections, biddings to downgrade any of them or exclude them from the second round. 'Perceptions affecting productivity' was ranked to be left out from the second round. However, we did not exclude this outcome as the outcome concept included measures such as work–life balance and job-satisfaction, which were brought forward as highly important by several stakeholders during the discussion. Therefore, it was upgraded for inclusion during the discussion.

### Stage 4: return to employment

The steering group discussed the disagreement on 'return to work—workers' perceptions, which were voted out by all stakeholder groups combined. The main argument against including it was whether the outcome actually measured work participation and whether it would be relevant to measure at the beginning and end of all types of interventions. However, this outcome was ranked as 'definite core outcome IN' by the employee representatives and some stakeholders from other groups were keen on including it in the second round. Therefore, it was not excluded. All versions of sickness absence outcomes received low ratings. Feedback from stakeholders suggested that this may be due to varying insurance schemes, how it is registered, lack of compensation of sick leave in many countries and in some cases punitive measures for being on registered sick leave. However, as there are interventions for which it is relevant to measure accumulated absence over time due to a health problem, and return to work may not be a relevant measure, the steering group decided to keep one absence outcome in the second round. We created a new outcome for the second round, with the aim of replacing sick leave

**Table 4** Results of the second-round ratings presented for all stakeholders combined (overall) and for each stakeholder group separately

| Outcome | Overall (n=55) | Researcher (n=37) | Employee (n=10) | Occupational health professional (n=4) | Policy maker (n=3) | Employer (n=1) |
|---|---|---|---|---|---|---|
| Clarifying text for new outcomes added to the second round (table 1 contains all initial outcomes) | ; % score 7–9 | ; % score 7–9 | ; % score 7–9 | ; % score 7–9 | ; % score 7–9 | score |
| Stage 1: Initiating employment | | | | | | |
| Any type of employment including self-employment (Outcome replaces all previous outcomes for stage 1. Also included in stage 2.) | 6.2; 55% | 6.6; 65% | 6; 40% | 5.3; 25% | 6; 33% | 1 n.a. |
| Stage 2: Having employment | | | | | | |
| Employment—any type of employment including self-employment | 7.9; 87% | 8; 89% | 7.4; 80% | 7; 7.5% | 8.7; 100% | 9 n.a. |
| Sustainable employment (New outcome based on suggestion from Delphi round 1 and the discussion: Any type of employment including self-employment AND one or more indicators of quality of working life (such as work life balance, healthy work, optimal work, work accommodations, decent work)) | 7.2; 66% | 6.7; 60% | 8.6; 100% | 7.8; 75% | 5; 0% | 9 n.a. |
| Work disability—self reported inability to engage in any work participation (Rephrased outcome based on the discussion). | 6.9; 64% | 7; 65% | 7.4; 80% | 7.8; 75% | 4.3; 0% | 1 n.a. |
| Employment—having lost employment | 6.5; 56% | 6.7; 60% | 7.6; 80% | 6; 25% | 4.3; 0% | 1 n.a. |
| Stage 3: Increasing or maintaining employment | | | | | | |
| Work productivity loss (economic evaluation) | 6.4; 62% | 6.6; 70% | 6.5; 50% | 5; 25% | 6.7; 67% | 1 n.a. |
| Work ability | 6.5; 60% | 6.6; 60% | 7.3; 90% | 5.5; 25% | 6.7; 67% | 1 n.a. |
| Work activity impairment | 6; 51% | 5.5; 41% | 7.5; 80% | 6.8; 50% | 6; 67% | 9 n.a. |
| Perceptions affecting productivity | 5.2; 35% | 4.7; 24% | 6.8; 70% | 5; 50% | 4.3; 0% | 9 n.a. |
| Stage 4: return to employment | | | | | | |
| Return to employment | | | | | | |
| Return to work—proportion of workers | 8; 91% | 8; 92% | 7.6; 90% | 8.5; 100% | 7.7; 67% | 9 n.a. |
| Time to return to work | 7.8; 80% | 8; 84% | 7.6; 80% | 7.8; 75% | 6; 33% | 9 n.a. |
| Sustainable return to work—proportion of workers | 7.2; 62% | 7.2; 62% | 7.7; 70% | 7.8; 75% | 5.3; 33% | 9 n.a. |
| Duration of absence from work due to health problem over a period of time. (This outcome replaces previous sickness absence outcomes based on the suggestions from round 1 and the discussion) | 6; 38% | 6; 40% | 6.7; 40% | 5.5; 25% | 6.7; 33% | 1 n.a. |
| Return to work—the worker's perceptions | 5.4; 38% | 5.1; 24% | 7.4; 90% | 3.5; 25% | 4.3; 33% | 9 n.a. |

Ratings are presented for the four work participation stages. The outcomes from the Delphi rounds are presented within each stage from most to least highly rated. Only outcomes which received a mean score of 7–9 for 50%–80% of the participants in the first Delphi round are included. We present the mean score ($\bar{X}$) given to each outcome per group and the percentage of the stakeholders within each group who rated an outcome with a score between 7 and 9. Further description of analysis can be found in section the 'Data analysis'.
N.a, not applicable.

terminology: 'duration of absence from work due to a health problem over a period of time'. In addition, we added one outcome 'having lost employment' to this stage, as an indicator of workers who do not return to work at all after a period of health-related absence.

**Second Delphi round: final vote on core outcomes**
Results of the second round are presented in tables 4 and 5 and contains the COS for Work.

**Stage 1: initiating employment**
'Having any type of employment including self-employment' as an outcome for interventions aiming

to help unemployed people get work did not meet the consensus criteria to be included in the COS for Work.

**Stage 2: having employment**
'Any type of employment including self-employment' received 87% consensus and was therefore included in the COS for Work

**Stage 3: increasing or maintaining productivity at work**
None of the outcomes in this stage received enough consensus to be included in the COS for Work. However, 'work productivity loss', 'work ability' and 'work activity impairment' received similar ratings to round one

**Table 5** COS for work

| Stage of work participation | COS for work |
| --- | --- |
| Interventions and study populations for which the outcomes are relevant within the stage | Outcomes which should always be measured in intervention studies relevant for the stage of work participation |
| Having employment<br><br>Relevant for any intervention and any type of health problem. Any type of health condition may impair work participation at the most elemental level, having work | 1. Having any type of employment including self-employment |
| Return to work<br><br>Interventions expected to help people who are absent from work due to a health problem return to work | 1. Proportion of workers that return to work after being absent because of illness<br>2. Time to return to work in workers absent because of illness |
| Outcomes which were selected by the panel to always be measured in studies measuring an effect on work participation.<br>COS, core outcome set. | |

indicating they are seen as important but not critical for the COS for Work.

### Stage 4: return to employment

Two outcomes received overall consensus for inclusion: (1) 'return to work—proportion' and (2) 'time to return to work'. 'Proportion of workers with sustainable return to work' was voted as important but not critical by all stakeholders. 'Workers' perception on return to work' received 90% agreement within the employee representatives group for inclusion but very low scores by other stakeholder groups. The 'duration of absence from work due to health problem over a period of time' also received low scores for inclusion (38% score between 7 and 9), making it a higher rating than all the sick leave outcomes from the first round—but nonetheless excluding any absenteeism outcome from consideration for the COS for Work.

### DISCUSSION
### Summary of main findings

Our international Delphi panel achieved consensus on three work participation outcomes for inclusion in COS for Work. As a minimum, all intervention studies addressing work participation should include 'any type of employment including self-employment' as an outcome. Intervention studies which include participants who are absent from work due to a health problem should include the outcomes 'proportion of workers that return to work after being absent because of illness' and 'time to return to work'.

### Strengths

We followed recommendations by the COMET initiative[5] to design the Delphi study, as well as the COS-STAD[9] and the COS-STAR.[10] For example, based on these standards we described the consensus definition a priori in a publicly available protocol. In addition, we used clearly communicated and specifically designed criteria

for deciding which additional outcomes suggested by stakeholders would be suitable for COS for Work. Our SR provided us with a comprehensive overview of how work participation is measured and reported internationally. We were able to cluster all the outcome concepts we found in the literature into 24 outcomes. Working with a relatively low number of outcomes is a facilitating factor for having focused discussions with stakeholders and may have contributed to the low attrition rate.[11 12]

The DelphiManager software was designed by the COMET initiative to conduct Delphi studies.

We had a group of international stakeholders representing a wide range of expertise relevant for research in the field of OH. The online discussions provided valuable exchange of perspectives between the stakeholders. For instance, through sharing the information on the various social security systems it became clear why some outcomes would be less feasible to measure internationally.

### Limitations

A limitation of the study is that we do not have core outcomes for all four work-participation stages. However, for some outcomes that did not meet the inclusion criterion, the consensus was still high and they are good candidates to be considered for measurement in studies (see table 6).

Another limitation of this study is that we had difficultly recruiting employee and employer representatives. The employee group did not join the first online discussion. As our panel consisted mostly of researchers it is likely that views of the other four stakeholder groups were underrepresented. Nonetheless, as COS for Work is developed to be used by researchers we believe that it addresses issues researchers face at an elemental level. As we presented the work participation stages with relevant outcomes in order of most to least commonly measured[6] the 'initiating employment' stage might have received less, and possibly not enough, attention.

**Table 6** Candidate outcomes

| Stage of work participation—interventions and study populations for which outcomes are relevant within the stage | Potential (additional) future COS outcomes which did not meet the cut-point for COS for Work inclusion |
|---|---|
| Stage 1: Initiating employment<br>Interventions aiming to help unemployed people with distance to the job market due to a health problem get work | 1. Having any type of employment including self-employment |
| Stage 2: Having employment<br>Relevant for any intervention and any type of health problem. Any type of health condition may impair work participation at the most elemental level, having work | 1. Sustainable employment<br>2. Work disability—self reported inability to engage in any work participation<br>3. Employment—having lost employment |
| Stage 3: Increasing or maintaining productivity at work<br>Interventions expected to help workers who are not on sick leave to maintain or increase productivity at work despite a health problem | 1. Work productivity loss (economic)<br>2. Work ability<br>3. Work activity impairment |
| Stage 4: Return to work<br>Interventions expected to help people who are on sick leave return to work | 1. Sustainable return to work—proportion of workers<br>2. Workers' perceptions on return to work<br>3. Having lost employment |
| These outcomes were also high ranking (by at least one group) and need further investigation in which context they could be (additional) COS for work outcomes.<br>COS, core outcome set. | |

Most participants were from Europe and North America. This might have influenced which outcomes are seen as most relevant for studies conducted on these continents and the results might be less applicable to low-income and middle-income countries.

## Results compared with previous findings

In terms of defining core outcomes for work participation most research to date has been done by the Outcome Measures in Rheumatology initiative.[13] Their main focus is on worker productivity loss including outcomes such as absenteeism, presenteeism and costs, and focus on people with rheumatic diseases and musculoskeletal diseases.[14 15] We consider that COS for Work is a suitable addition to include in studies which measure such productivity outcomes.

Sustainable employment was proposed as a potential COS outcome by a stakeholder during the first Delphi round. While we did not find this outcome as a commonly measured work outcome in the literature[6] it appears to be a highly important outcome from the employee perspective. Sustainable return to work is commonly measured in terms of having the ability to work for an extended period. However, employee representatives and some researchers understand sustainable employment in terms of quality of work, having decent work and a good work-life balance. Defining sustainability in terms of healthy work is in line with results of a study done among employees on how they perceive sustainable work.[16]

## Implications for practice and research

In table 6, we present the candidate outcomes that did not meet the cut-off but were also highly ranked by

stakeholders. These findings are not reported in the results section as the aim of the study was to determine core outcomes for work participation. However, there were several outcomes for which there was strong disagreement between the stakeholder groups. The reasons are discussed in the results section; the summary of the stakeholder discussions and ratings of the Delphi voting rounds. It is possible that the candidate outcomes did not reach the consensus criterion due to the background of the researchers (most researchers do not investigate outcomes relevant to help people with initiating employment). In addition, group discussion generally supported an outcome such as 'sustainable employment', but lack of consensus on the exact definition may have influenced low scores in the second Delphi voting round. Further research is needed to investigate which of these outcomes should be prioritised under which circumstances.

In our next study we will define and evaluate how to best measure 'having any employment' and 'return to work' in the COS for Work. Use of COS for Work by researchers will greatly help with evidence synthesis for researchers thereby helping decision making for practice and policy.

**Author affiliations**
[1]Department of Public and Occupational Health, Coronel Institute of Occupational Health, Amsterdam Public Health Research Institute, Cochrane Work, Amsterdam UMC Location AMC, University of Amsterdam, Amsterdam, The Netherlands
[2]Department Epidemiology and Data Science, Amsterdam Public Health Research Institute, Amsterdam UMC Location AMC, University of Amsterdam, Amsterdam, The Netherlands
[3]Guy's and St Thomas' NHS Foundation Trust, London, UK
[4]King's College London Faculty of Life Sciences and Medicine, London, UK
[5]Centre for Epidemiology Versus Arthritis, Faculty of Biology, Medicine and Health, University of Manchester, Manchester, UK

⁶Manchester University NHS Foundation Trust, NIHR Manchester Biomedical Research Centre, Manchester, UK

⁷MRC Versus Arthritis Centre for Musculoskeletal Health and Work, University of Southampton, Southampton, UK

⁸Research Unit EbIM, Evidence Based Insurance Medicine, Division of Clinical Epidemiology, University Hospital Basel, Basel, Switzerland

⁹Department of Public and Occupational Health, Coronel Institute of Occupational Health, Amsterdam UMC Location AMC, University of Amsterdam, Amsterdam, The Netherlands

**Acknowledgements** We would like to thank all the stakeholders who participated in this study for contributing to the results for COS for Work, the investment of time and the lively discussions.

**Collaborators** Delphi participants: Aasdahl, Lene (NO); Anema, Johannes (NL); Berkowitz, Debbie (US); Bethge, Matthias (DE); Bühler, Jonas (CTJH); Bültmann, Ute (NL); Böckerman, Petri (FI); Coons, Trevor (US); Curti, Stefania (IT); Crosse, Caroline (AU); de Boer, Angela (NL); Dorstyn, Diana (AU); Foster, Nadine (AU); Friberg, Emilie (SE); Gehanno, Jean Francois (FR); Godderis, Lode (BE); Goodson, Nicola (UK); Graveline, Christine (CA); Gross, Douglas (CA); Hara, Karen Walseth (NO); Hannu, Timo (FI); Hoff, Andreas (DK); Hensing, Gunnel (SE); Hegewald, Janice (DE); Hoorntje, Alexander (NL); Janssen, Svenja (NL); Lacaille, Diane (CA); Lam, Raymond (CA); Landsbergis, Paul (US); Luiza Comper, Maria (BR); Lytsy, Per (SE); Macfarlane, Gary (UK); Mantis, Steve (CA); Merry, Kohle (CA); Meyers, Alysha (US); Neupane, Subas (FI); Nygård, Clas-Hakan (FI); Oyeflaten, Irene Larsen (NO); Pingle, Shyam (IN); Prior, Yeliz (UK); Schaafsma, Frederieke (NL); Samant, Yogindra (NO); Shaw, William (US); Snyder, Alexis (US); Steenstra, Ivan (CA); Sturkenboom, Ingrid (NL); Suijkerbuijk, Yvonne (NL); Tsutsumi, Akizumi (JP); Urquhart, Donna (AU); van Ee, Ilse (NL); van Zaanen, Yvonne (NL); Yamaguchi, Sosei (JP); Zhang, Wei (CA); Walker Bone, Karen (UK).

**Contributors** MR, JH, JHV, RK, CTJH, ML, IM and SV contributed to the conception, the study design and the execution of the Delphi. MR and JH tested the DelphiManager software prior to the first Delphi round. MR and JHV analysed the data. MR and JH assessed data quality. MR drafted the first version of the manuscript and the minor revisions, and JH, JHV, RK, CTJH, ML, IM and SV contributed to the review and the editing process. JH is the guarantor of the publication.

**Funding** This research received no specific grant from any funding agency in the public, commercial or not-for-profit sectors. SV is supported by Versus Arthritis (grant numbers 20385, 20380) and the NIHR Manchester Biomedical Research Centre.

**Competing interests** None declared.

**Patient and public involvement** Patients and/or the public were involved in the design, or conduct, or reporting, or dissemination plans of this research. Refer to the Methods section for further details.

**Patient consent for publication** Not applicable.

**Ethics approval** The research was conducted in accordance with the Declaration of Helsinki. The research proposal was submitted and approved by the Medical Ethical Committee of the Amsterdam UMC, Academic Medical Centre, Amsterdam, which judged that a comprehensive evaluation was not required since this study was not subject to the Netherlands' Medical Research Involving Human Subjects Act (reference: W22_129 # 22.171). Participants gave informed consent to participate in the study before taking part.

**Provenance and peer review** Not commissioned; externally peer reviewed.

**Data availability statement** All data relevant to the study are included in the article or uploaded as online supplemental information.

**ORCID iD**
Margarita Ravinskaya http://orcid.org/0000-0003-4280-8887

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
