## [Reviewer comments · BMJ Open]

ARTICLE DETAILS

TITLE (PROVISIONAL)	Which outcomes should always be measured in intervention studies for improving work participation for people with a health problem? An international multi-stakeholder Delphi study to develop a core outcome set for work participation (COS for Work)
AUTHORS	Ravinskaya, Margarita; Verbeek, Jos; Langendam, Miranda; MADAN, IRA; Verstappen, Suzanne; Kunz, Regina; Hulshof, C.; Hoving, Jan

VERSION 1 – REVIEW

REVIEWER	Lachman, Peter Royal College of Physicians of Ireland, Quality
REVIEW RETURNED	04-Nov-2022

GENERAL COMMENTS	A well constructed Delphi yet the main challenge is the skew to researchers and only 1 employer. Probably the employer view should be considered as superfluous to the study as it is only one person's view and may not represent the view of employers in general. Another example of the potential bias as to what is important is in the assessment that Two outcomes received low overall ratings but were ranked as “definite core outcome” by the employees: “workers’ perceptions on return to work” and “having lost employment” in the return to work stage. While you recommend further research is needed perhaps this is more indicative of the researcher bias in the make up of the respondents. If one is to consider the lived experience of the employee and the lived experience of the employer perhaps there would be a different outcome if you had a different ratio rather than the 60% researcher and only 17% employee and only 1% employer. You discuss this in limitations - though it could be stronger; if this is purely a research tool, then it may not be important. Finally it could be useful to consider how the research relates to the impact of COVID on workforce, of long COVID on all 4 stages - the framework could be a very useful way to approach the challenges
--

REVIEWER	Chaar, Betty Faculty of Pharmacy, World Hospital Pharmacy Research Consortium, School of Pharmacy
REVIEW RETURNED	11-Nov-2022

GENERAL COMMENTS

Thank you for the opportunity to review this interesting manuscript. It is an interesting topic and has potential for future practice guidelines.

I accepted to review this paper not because I am an OT by training, but for my interest in the methodology and outcomes extracted. So, what I have to comment about is from the perspective of an external reader, and from that perspective it was not an easy read to follow. As BMJ Open is not a profession-specific journal, then there needs to be some care involved in setting the scene and explaining exactly where your results fit in as outcomes that will inform practice at some stage in the future. I daresay, even for OT researchers, this may be a complex paper to follow. Not for lack of rigor or accuracy, but for the narrative and the explanation of the contextual significance of this paper and its content to a whole profession across the world.

If I may, I suggest at the very beginning, to clearly set the scene about the core outcome measures and where they come in to play. Explain what "interventions" you are referring to. You intermittently mention health or ill health or even disability. Where does ill health fit exactly in your narrative? I am assuming that OT intervention is about improving health, in many ways. Try to describe what intervention or what level of intervention you were investigating core measures for (or were you referring to intervention at any level? in which case that might have been better explained). I was not able to figure out what you meant precisely, and it might have also helped in the Delphi method applied, to further clarify what level of intervention or kind of intervention you referred to in your items for discussion.

I was also curious about what age group/s you may have been interested in. Are these outcomes applicable to all interventions and all ages? Can they be generalized?

These points may be simply given to you as authors in this field, but for a reader interested in methods and outcomes, they are issues that might have enhanced the readability of the manuscript.

Sampling: Was there a reason for your choice of participant categories? It would be interesting to know how you were able to recruit international participants. It would also be helpful if the difference between "employee" and other categories such as Occupational Health professionals was explained. Are employees not professionals?

Headings in Table 3 could be better presented.

Table 4 needs tidying up. The mean score column is not straight forward.

Discussion: the reader anticipates a little more contextualization of results and perhaps significance of the findings. In the section 'implications for practice and research' Table 6 is presented. You might wish to move this table outside the discussion and report with results. It is not quite fitting here as authors do not actually discuss (or at least hypothesize) why two outcomes were rated low but ranked 'definitely core outcomes'. Were these items not discussed in the interviews? or inadequately explained by participants? or was there an unexplained difference between the sample of participants and who they represented? This phenomenon needed to be clarified.

	It would be nice to have countries of acknowledged participants in brackets with their names (this is just a suggestion). Overall: with some thought given to further clarity in this manuscript I believe it is publishable. Avoid jargon and obscure terminology. Clarity of statements and how they can be applicable is important, to allow for universal adoption of outcome measures found, which is the ultimate goal.
--	--

REVIEWER	Edgelow, Megan Queen's University
REVIEW RETURNED	08-Dec-2022

GENERAL COMMENTS	Thank you for this high quality manuscript. Your study had excellent fidelity to the Delphi method and is very clearly written. Thank you for this important work. I do not have any revisions to suggest.
--

VERSION 1 – AUTHOR RESPONSE

Reviewer: 1

Dr. Peter Lachman, Royal College of Physicians of Ireland Comments to the Author:

A well constructed Delphi yet the main challenge is the skew to researchers and only 1 employer. Probably the employer view should be considered as superfluous to the study as it is only one person's view and may not represent the view of employers in general.

Another example of the potential bias as to what is important is in the assessment that Two outcomes received low overall ratings but were ranked as “definite core outcome” by the employees: “workers’ perceptions on return to work” and “having lost employment” in the return to work stage.  While you recommend further research is needed perhaps this is more indicative of the researcher bias in the make up of the respondents. If one is to consider the lived experience of the employee and the lived experience of the employer perhaps there would be a different outcome if you had a different ratio rather than the 60% researcher and only 17% employee and only 1% employer.

You discuss this in limitations - though it could be stronger; if this is purely a research tool, then it may not be important.

Reply: Dear. Dr Lachman, thank you for your feedback. We agree that it is unfortunate that we were only able to recruit one employer representative. We contacted several organizations for representatives such as Business Europe, but without success. COS for Work is developed for use by researchers as is described in the introduction: “*Our aim was to develop a generic COS for Work to be used in intervention studies which expect to have an effect on work participation.*” (line 81-82) and in the discussion: “*Another limitation of this study is that we had difficulty recruiting employee and employer representatives. The employee group did not join the first online discussion. As our panel consisted mostly of researchers it is likely that views of the other four stakeholder groups were underrepresented. Nonetheless, as COS for Work is developed to be used by researchers we believe that it addresses issues researchers face at an elemental level*” (lines 370-373). Nonetheless, all relevant stakeholders should be involved in COS for research purposes, therefore we did try to

include as many as possible. We hope that we describe with sufficient clarity that COS for Work is intended for research purposes and therefore it is not a debilitating limitation that the group consisted mostly of researchers.

Finally it could be useful to consider how the research relates to the impact of COVID on workforce, of long COVID on all 4 stages - the framework could be a very useful way to approach the challenges.

Reply: Thank you for this observation. We hope that researchers will measure the COS for Work outcomes as a minimum and possibly use the framework for selection of additional outcomes which are relevant for the health problem evaluated in their research project. Following your suggestion, we added use of COS for Work for the evaluation of long COVID research (line 87):

“Our aim was to develop a generic COS for Work to be used in intervention studies which expect to have an effect on work participation. As our COS is generic, and relevant to different diseases, it can be used for any type of intervention. For example, pharmaceutical studies for people with Rheumatoid Arthritis may not only improve clinical outcomes but indirectly also help participants participating in work. Such studies mostly measure work outcomes as a secondary outcome. Other types of interventions aim directly in helping people with work participation, such as return to work interventions for cancer survivors or workers with long COVID.”

Reviewer: 2

Dr. Betty Chaar, Faculty of Pharmacy, World Hospital Pharmacy Research Consortium Comments to the Author:

Thank you for the opportunity to review this interesting manuscript. It is an interesting topic and has potential for future practice guidelines.

Reply: Thank you Dr. Chaar for taking the time to provide us with considerate feedback which will serve better understanding of our work for the general public.

I accepted to review this paper not because I am an OT by training, but for my interest in the methodology and outcomes extracted. So, what I have to comment about is from the perspective of an external reader, and from that perspective it was not an easy read to follow. As BMJ Open is not a profession-specific journal, then there needs to be some care involved in setting the scene and explaining exactly where your results fit in as outcomes that will inform practice at some stage in the future. I daresay, even for OT researchers, this may be a complex paper to follow. Not for lack of rigor or accuracy, but for the narrative and the explanation of the contextual significance of this paper and its content to a whole profession across the world.

If I may, I suggest at the very beginning, to clearly set the scene about the core outcome measures and where they come in to play. Explain what "interventions" you are referring to. You intermittently mention health or ill health or even disability. Where does ill health fit exactly in your narrative? I am assuming that OT intervention is about improving health, in many ways. Try to describe what intervention or what level of intervention you were investigating core measures for (or were you referring to intervention at any level? in which case that might have been better explained). I was not able to figure out what you meant precisely, and it might have also helped in the Delphi method applied, to further clarify what level of intervention or kind of intervention you referred to in your items for discussion.

I was also curious about what age group/s you may have been interested in. Are these outcomes applicable to all interventions and all ages? Can they be generalized?

Reply: Following the suggestion, we have added the following clarification at the beginning of the introduction:

“Our aim was to develop a generic COS for Work to be used in intervention studies which expect to have an effect on work participation. As our COS is generic, and relevant to different diseases, it can be used for any type of intervention. For example, pharmaceutical studies for people with Rheumatoid Arthritis may not only improve clinical outcomes but indirectly also help participants participating in work. Such studies mostly measure work outcomes as a secondary outcome. Other types of interventions aim directly in helping people with work participation, such as return to work interventions for cancer survivors or workers with long COVID.” (lines 81-87).

These points may be simply given to you as authors in this field, but for a reader interested in methods and outcomes, they are issues that might have enhanced the readability of the manuscript.

Sampling: Was there a reason for your choice of participant categories?

Reply: In lines 138-146 we have elaborated on whom the stakeholders represented and why they were selected as key stakeholders for COS for Work:

1. *OH researchers – experienced researchers in OH field from varying disciplines, such as: health economy, occupational mental health, insurance medicine, workplace interventions, RtW interventions*
2. *OH professionals – physiotherapists, orthopedic surgeon, rehabilitation specialist*
3. *Workers/employees and patients – stakeholders representing patients and workers as members of patients federations, workers alliances and labor unions*
4. *Policy makers – stakeholders working for organizations creating OH policies such as the National Institute for Insurance against Accidents at Work, EU-OSHA,*
5. *Employer – OH case manager*

These five stakeholder groups were chosen as key stakeholders. Researchers will use the COS for Work and have experience with work participation outcomes measurement and the remaining stakeholder groups will use the results of what is researched in their professional or daily life.

It would be interesting to know how you were able to recruit international participants.

Reply: We asked participants who completed a survey we conducted prior to the Delphi study and contacted participants who indicated that they would be interested in participating with further studies on the development of the core outcome set.

We recruited most stakeholders by contacting them from an international survey which we did prior to the Delphi study. This has been specified in the methods section (line 150).

“Stakeholders were recruited from our phase two international survey (6) and our professional network.”

It would also be helpful if the difference between "employee" and other categories such as Occupational Health professionals was explained. Are employees not professionals?

Reply: This was indeed not clearly explained, however the elaboration in lines 138-146 should clarify the difference.

Headings in Table 3 could be better presented.

Reply: We have adjusted the table description as follows (lines 234-236):

“Characteristics of survey respondents for two Delphi rounds (n; %). In total 58 participants registered as participants in Delphi Manager. Each participant belonged to one stakeholder group. In every round three participants did not complete the voting.”

In addition, we added “number of participants” beneath “Delphi round 1 and 2” headings to make it clearer that the presented numbers are about the number of participants per stakeholder group who participated in each Delphi round.

Table 4 needs tidying up. The mean score column is not straight forward.

Reply: We hope that an elaborated explanation of the table results makes the mean score columns more straight forward (lines 324-330):

“Results of the second-round ratings presented for all stakeholders combined (overall) and for each stakeholder group separately. Ratings are presented for the four work participation stages. The outcomes from the Delphi rounds are presented within each stage from most to least highly rated. Only outcomes which received a mean score of 7-9 for 50% - 80% of the participants in the first Delphi round are included. We present the mean score () given to each outcome per group and the percentage of the stakeholders within each group who rated an outcome with a score between 7-9. Further description of analysis can be found in section ‘data analysis’. N.a.: not applicable.”

Discussion: the reader anticipates a little more contextualization of results and perhaps significance of the findings. In the section 'implications for practice and research' Table 6 is presented. You might wish to move this table outside the discussion and report with results. It is not quite fitting here as authors do not actually discuss (or at least hypothesize) why two outcomes were rated low but ranked 'definitely core outcomes'. Were these items not discussed in the interviews? or inadequately explained by participants? or was there an unexplained difference between the sample of participants and who they represented? This phenomenon needed to be clarified.

Reply: We agree that it is not standard to present a table with studies' findings in the discussion section. Our team concluded that these results however did not fit within the results section of this manuscript because these are outcomes which are deemed important but did not meet the predefined consensus definition. We hope that the following clarification helps interpretation of these findings (lines 394-414)

“In table 6 we present the candidate outcomes that did not meet the cut-off but were also highly ranked by stakeholders. These findings are not reported in the results section as the aim of the study was to determine core outcomes for work participation. However, there were several outcomes for which there was strong disagreement between the stakeholder groups. The reasons are discussed in the results section; the summary of the stakeholder discussions and ratings of the Delphi voting rounds. It is possible that the candidate outcomes did not reach the consensus criterion due to the background of the researchers (most researchers do not investigate outcomes relevant to help people with initiating employment). In addition, group discussion generally supported an outcome such as ‘sustainable employment’, but a lack of consensus on the exact definition may have influenced low scores in the second Delphi voting round. Further research is needed to investigate which of these outcomes should be prioritized under which circumstances.”

It would be nice to have countries of acknowledged participants in brackets with their names (this is just a suggestion).

Reply: Thank you for the suggestion. We have added the countries of the participants in brackets behind their names (lines 433-443).

Overall: with some thought given to further clarity in this manuscript I believe it is publishable. Avoid jargon and obscure terminology. Clarity of statements and how they can be applicable is important, to allow for universal adoption of outcome measures found, which is the ultimate goal.

Reviewer: 3

Dr. Megan Edgelow, Queen's University

Comments to the Author:

Thank you for this high quality manuscript. Your study had excellent fidelity to the Delphi method and is very clearly written. Thank you for this important work. I do not have any revisions to suggest.

Dear Dr. Edgelow,

Thank you kindly for taking the time to review our manuscript and for such a complimentary reply.

VERSION 2 – REVIEW

REVIEWER	Lachman, Peter Royal College of Physicians of Ireland, Quality
REVIEW RETURNED	29-Jan-2023
GENERAL COMMENTS	Thank you for addressing the comments made which have clarified and improved your paper